# PchE Regulation of *Escherichia coli* O157:H7 Flagella, Controlling the Transition to Host Cell Attachment

**DOI:** 10.3390/ijms21134592

**Published:** 2020-06-28

**Authors:** Elisa Andreozzi, Gaylen A. Uhlich

**Affiliations:** Molecular Characterization of Foodborne Pathogens Research Unit, Eastern Regional Research Center, Agricultural Research Service, United States Department of Agriculture, Wyndmoor, PA 19038, USA; elisa.andreozzi@uniurb.it

**Keywords:** *Escherichia coli*, O157:H7, biofilm, cell adhesion, motility, flagella, *pchE*

## Abstract

Shiga toxins and intimate adhesion controlled by the locus of enterocyte effacement are major enterohemorrhagic *Escherichia coli* (EHEC) virulence factors. Curli fimbriae also contribute to cell adhesion and are essential biofilm components. The transcriptional regulator PchE represses the expression of curli and their adhesion to HEp-2 cells. Past studies indicate that *pchE* also represses additional adhesins that contribute to HEp-2 cell attachment. In this study, we tested for *pchE* regulation of several tissue adhesins and their regulators. Three adhesin-encoding genes (*eae*, *lpfA1*, *fliC*) and four master regulators (*csgD*, *stpA*, *ler*, *flhDC*) were controlled by *pchE*. *pchE* over-expression strongly up-regulated *fliC* but the marked flagella induction reduced the attachment of O157:H7 clinical isolate PA20 to HEp-2 cells, indicating that flagella were blocking cell attachments rather than functioning as an adhesin. Chemotaxis, motor, structural, and regulatory genes in the flagellar operons were all increased by *pchE* expression, as was PA20 motility. This study identifies new members in the *pchE* regulon and shows that *pchE* stimulates flagellar motility while repressing cell adhesion, likely to support EHEC movement to the intestinal surface early in infection. However, induced or inappropriate *pchE*-dependent flagellar expression could block cell attachments later during disease progression.

## 1. Introduction

*Escherichia coli* diarrheal disease in humans can be initiated by different pathotypes, each expressing distinct sets of virulence factors that adapt each pathotype for a specific niche and contribute to a unique disease syndrome. Six common diarrheal pathotypes have been described and extensively reviewed [1]: enterotoxigenic *E. coli* (ETEC), enteropathogenic *E. coli* (EPEC), enteroinvasive *E. coli* (EIEC), diffusely adherent *E. coli* (DAEC), enteroaggregative *E. coli* (EAEC), and enterohemorrhagic *E. coli* (EHEC). Certain virulence factors remain associated predominately with one specific pathotype but can occasionally transfer to a different pathotype resulting in strains that are poorly adapted to the human host and highly virulent. For instance, prophage-encoded Shiga toxins are the key virulence factor of EHEC [1]. More than 200 serotypes of *E. coli* produce Shiga toxin, of which O157:H7 is the serotype most often associated with severe disease [2]. However, the transfer of Shiga toxin to a serotype O104:H4 EAEC strain produced a highly pathogenic strain responsible for a large food-associated outbreak in Germany in 2012 [3]. The hybrid strain, which expressed both Shiga toxin and the strong adhesive and aggregative properties of EAEC, not only demonstrated the ongoing evolution of the pathotypes, but also emphasized the importance of strong adhesive phenotypes and biofilm formation for EHEC virulence [4].

Human clinical EHEC infections typically produce colitis that results in abdominal pain and diarrhea. Following the release of Shiga toxins, the colitis becomes hemorrhagic and can, in some instances, progress to systemic syndromes such as thrombotic thrombocytopenic purpura and diarrheal-associated hemolytic uremic syndrome (HUS). Such syndromes are characterized by vascular endothelial damage in various organs such as the kidneys, nervous system, and lungs [2,5].

While the production of Shiga toxin(s) is considered essential for disease, certain other factors, if not essential, maintain a high prevalence in those strains responsible for severe disease or large outbreaks. One such factor is the locus of enterocyte effacement (LEE) pathogenicity island [6]. The LEE encompasses five major operons and several smaller operons or individual genes that encode, among other elements, a type III secretion system, various secreted effector proteins, and several transcriptional regulators [7]. The global *E. coli* gene repressor, H-NS, prevents LEE expression until permissive conditions allow activation by Ler, a DNA-binding transcription factor encoded in LEE operon 1 [8]. Ler and H-NS have structural similarities in their DNA-binding domains, which allows for the control of LEE by competitive binding at shared target sites [9]. The regulation of *ler* is complex and has been covered in different reviews [10,11]. Essential transcriptional regulators controlling *ler* include the *grlRA* genes encoded in LEE and certain members of a family of five homologues (*pchABCDE*) of the *perC* gene of EPEC [12,13]. Members of the *pch* family are located in various horizontally transferred regions (HTR) outside the LEE region [14].

The colonization of the gastrointestinal tract (GIT) is an essential step in EHEC pathogenesis and a variety of proven and putative bacterial adhesins have been identified. One well-characterized protein is LEE-encoded intimin (the *eae* gene), which binds enterocytes expressing the translocated intimin receptor (Tir), leading to the intimate adhesion central to EHEC attaching and effacing lesions [6,15]. However, initial weaker contacts that link EHEC-expressing *espA* to enterocytes are required prior to the translocation of TIR and the establishment of strong intimate enterocyte attachments [16]. Intimin can also form weaker contacts with the host cell proteins nucleolin and β1 integrin that likely serve as preliminary attachments preceding intimate adhesion [17]. Colonization, hemorrhagic colitis, and HUS have also been reported in O157:H7 strains that do not carry the LEE operon and therefore depend on attachment factors outside of the LEE operons. Factors other than intimin that might be important in GIT attachment and colonization by EHEC have been summarized in different reports [18,19,20,21].

Among other non-intimin attachment factors, long polar fimbriae (Lpf) [22,23], the hemorrhagic coli pilus (HCP) [24,25] *E. coli* common pilus (ECP) [26], F9 fimbriae [27], and *E. coli* laminin-binding fimbriae (ELF) [28] have all been shown to participate in *E. coli* adhesion to cultured human cells or aid the colonization of human or animal hosts. In addition, *E. coli* type 1 fimbriae (*fimBEACDFGH* operon) attach to bovine rumen epithelia [29], although in serotype O157:H7 the reversible 16-bp promoter element controlling expression was found to be in the “off” orientation in the O157:H7 strains investigated thus far [30].

Flagella play a role in the colonization of animal GIT by forming attachments with intestinal epithelial cells, but their complete role in colonization of the human GIT is unclear [31,32,33]. Flagellar motility is a complex system in *E. coli* and the 14 flagellar operons involved are controlled by a regulatory cascade involving three classes of temporally expressed genes where the expression of class 2 and class 3 members is dependent on gene expression from the previous classes [34,35]. Class 1 contains the master regulators *flhD* and *flhC,* which activate class 2 genes. Class 2 includes genes for the components of the rotary motor and the FliA sigma factor (sigma 28) controlling class 3 genes. Class 3 genes include structural components of the flagella such as flagellin (*fliC*) and the chemotaxis genes.

Curli fimbriae are well-characterized adhesive organelles that are an essential component of O157:H7 biofilms. Curli are highly regulated and expressed at environmental temperatures using the RpoS sigma factor in conjunction with the global regulator CsgD and the MlrA transcription factor [36,37]. However, *csgD* is also expressed at 37 °C, by a mechanism that has yet to be clearly defined, and it can play a role in attachment to cultured tissue cells and animal GI tissues. A recent study showed that *csgD* and curli (*csgA*) were strongly downregulated by plasmid over-expression of *pchE* and that a steady natural level of *pchE* in O157:H7 was repressed by SOS induction following treatment with sulfamethoxazole/trimethoprim (SMX-TM) at 540/108 μg/L [38]. In that study, *pchE* repressed HEp-2 cell adhesion to lower levels than those observed following curli subunit deletion indicating that there may be additional adhesins in the *pchE* regulon. To date, natural inducers of *pchE* have yet to be identified.

The purpose of this study was to investigate whether the *pchE* gene controls important cellular adhesins previously described for serotype O157:H7 or certain master regulators involved in biotic and abiotic adhesion [39,40,41]. Quantitative real-time PCR (qRT-PCR)-derived gene expression changes in the adhesins (LpfA1, HCP, ECP, F9 fimbriae, ELF, flagella, curli, intimin) and global regulators (CsgD, Ler, H-NS, StpA, FlhCD) were compared in four different strains with various levels of *pchE* expression: 1) wild-type O157:H7 clinical isolate PA20; 2) PA20 with deletion of *pchE*; 3) PA20 expressing plasmid-cloned *pchE*; and 4) PA20 exposed to an SOS-inducing SMX-TM concentration. The significance of certain gene expression changes was tested in swimming motility and HEp-2 cell adhesion assays.

## 2. Results

### 2.1. PchE Regulates PA20 Adhesins 

Figure 1 shows the qRT-PCR-derived fold-change (FC) expression differences determined from comparisons of wild-type PA20 with PA20Δ*pchE* and PA20 SMX-TM. Although the conditions that induce *pchE* expression have not been identified, we also show the expression changes that might result from *pchE* induction, testing PA20 with expression of *pchE* from an inducible promoter (strain PA20 pSE380::*pchE*). Three adhesin genes (*eae*, *lpfA1*, and *fliC*) were differentially expressed (FC ≥ 2) following *pchE* over-expression (Figure 1A). The largest regulatory change was observed in PA20 *fliC* expression where pSE380::*pchE* resulted in a 20-fold increase. In contrast, the expression of the adhesin genes *lpfA1* and *eae* was modestly reduced (twofold and 2.7-fold, respectively). The *csgA* gene, which was strongly repressed by *pchE* over-expression at 37 °C in an earlier study, was reduced only 1.6-fold under these conditions testing Eagle’s Minimal Essential Medium (EMEM) broth rather than T-medium agar. The deletion of *pchE* had little effect on adhesin gene expression, with no gene registering an expression change that reached the twofold difference threshold, probably due to low *pchE* expression under the tested conditions. Therefore, these results identify *lpfA1*, *eae*, and *fliC* as potential new adhesin genes in the *pchE* regulon.

### 2.2. PchE Affects the Master Regulators of Biofilm Formation, Motility, and Virulence

*pchE* regulation of the CsgA subunit, involved in cell adhesion and biofilm formation, is mediated in part through an effect on the master curli regulator, CsgD. Therefore, we tested additional global or master regulators controlling biofilms or adhesion for a *pchE* regulatory effect.

Four regulators (*csgD*, *stpA*, *ler*, and *flhDC*) were changed (FC ≥ 2) by *pchE* over-expression. Over-expressed *pchE* induced a 3.1-fold reduction in *csgD*, lower than the 6.3-fold repression noted in our earlier study testing slightly different conditions (Figure 1A). The *ler* gene, which controls the LEE pathogenicity island and intimate adhesion, was induced 2.1-fold (Figure 1A). The *hns* gene, which opposes *ler* regulatory effects*,* was unchanged. However, *stpA*, encoding a protein similar to H-NS, was repressed 2.5-fold by *pchE* over-expression (Figure 1A). Finally, the class 1 transcription factors *flhC* and *flhD*, which directly or indirectly control the expression of the flagellar operons, were increased > 2-fold by *pchE* over-expression (Figure 1B).

Only two regulators were affected by *pchE* deletion using a FC ≥ 2 expression difference as the threshold. c*sgD* was repressed 2.1-fold compared to the controls and the two components of the *flhDC* complex were decreased 2.0 and 1.9, respectively (Figure 1B). 

### 2.3. PchE Regulates Multiple Flagellar Operons

Strong regulation of *fliC* by *pchE,* along with stimulation of the *flhDC* complex, prompted the testing of other genes in the *flhDC* regulon (Figure 1). Genes in the different flagellar operons, including chemotaxis, motor, structural, and regulatory genes, showed the greatest changes in expression magnitude in this study; all were increased by *pchE* expression (and unchanged by *pchE* deletion). Class 2, *fliC,* and class 3, *fliA*, encoding the flagellar subunit and the transcription factor required for transcription of class 3 genes, were increased from 12- to 24-fold. The class 3 motor and chemotaxis genes tested (*motB*, *cheA* and *cheR*) were also markedly enhanced, ranging from nearly five- to ninefold.

### 2.4. Induction by Antibiotics Has Minor Effects on Adhesins or Master Regulators

The effect of an SOS-inducing level of SMX-TM, which imposes a repressive effect on *pchE* expression, was also tested on genes in this study [42] (Figure 1). Only two genes exhibited differential expression changes that surpassed the FC ≥ 2-fold threshold. SMX-TM resulted in increased *csgA* expression (FC = 2.4), probably by relieving the *pchE* suppression of *csgD* noted here. The only other expression difference with a FC ≥ 2-fold following SMX-TM exposure was that for the F9fim gene, which was repressed 2.5-fold but affected little by *pchE* deletion. Apparently, SMX-TM repression of the F9fim gene is not mediated through *pchE*.

### 2.5. PchE Regulates PA20 Flagellar Motility

The expression studies here indicate that control of flagella motility may be a major function of PchE. Therefore, motility differences among PA20, PA20 pSE380::*pchE*, and PA20Δ*pchE* were compared using soft agar motility assays (Figure 2A). Plasmid over-expression of *pchE* increased the diameter of the PA20 motility zone by nearly 50% but the deletion of *pchE* resulted in little change in PA20 motility, results consistent with the qRT-PCR expression findings.

We also compared PA20 strains with deletion of either flagellar or curli genes, as curli fimbriae could also affect motility. Deletion of the flagellin gene, *fliC*, was sufficient to completely eliminate PA20 migration (Figure 2B). However, deletion of both the curli master regulator, *csgD,* and curli structural genes, *csgBA*, strongly reduced PA20 migration, indicating that curli may enhance motility by affecting either the expression or function of flagella (Figure 2B).

Finally, to confirm that the PA20 migration differences induced by the *pchE* plasmid were dependent on flagella, the motility of strains PA20 pSE380::*pchE* and PA20*ΔfliC* pSE380::*pchE* were compared. PA20 motility remained entirely *fliC*-dependent, even in the presence of *pchE* over-expression (Figure 3A).

### 2.6. Butyrate Augments pchE-induced Motility

Sodium butyrate enhances both flagella expression and motility in serotype O157:H7 by stimulating flagellar genes in all three classes, similar to the effects observed here following *pchE* overexpression; class 1 by affecting regulatory protein Lrp and classes 2 and 3 also by an Lrp-independent mechanism (Figure 1) [43]. When PA20 pSE380::*pchE* was compared with PA20 pSE380 in agar with and without 20 mM sodium butyrate, the optimum dose used in that study, both butyrate and the over-expression of *pchE* were capable of strongly enhancing PA20 motility (Figure 3B). Furthermore, an additive effect was seen when *pchE*, driven from an artificial promoter, was expressed in the presence of 20 mM butyrate.

### 2.7. PchE Is Repressed by SMX-TM but Unaffected by Butyrate

To determine whether the butyrate effect on motility could be mediated by PchE, in addition to Lrp, a *pchE* promoter fusion with *lacZ* was constructed and tested in PA20 (PA20Δ*pchE* pMLB1034::*pchE*) at 37 °C in Minimal Essential Medium (MEM). First, the promoter fusion was tested by PA20 exposure to SMX-TM. SMX-TM reduced PA20 *pchE* transcript counts > 2-fold in an earlier RNA-Seq study. When ß-galactosidase activity of PA20Δ*pchE* pMLB1034::*pchE* was followed over a 3 h time period, *pchE* promoter activity steadily increased to a maximum at 1 h before gradually declining (Figure 4A). When exposed to SMX-TM, the promoter activity was reduced compared to the control strain, differing slightly at 30 min post-exposure and reaching a maximum difference (threefold) by 60 min. Therefore, the promoter fusion replicated the expression trends observed in the RNA-seq studies, validating the effectiveness of the fusion for assaying *pchE* promoter activity. When compared in the presence of 20 mM sodium butyrate, *pchE* promoter activities of the exposed and non-exposed strains remained similar indicating that butyrate does not activate *pchE* under the tested conditions (Figure 4B).

### 2.8. Electron Microscopy

Flagella expression on bacterial strains was investigated using scanning electron microscopy (SEM). Plasmid pSE380 was maintained in PA20 to confirm that any observed flagellar expression in the presence of pSE380::*pchE* would be dependent on cloned *pchE* rather than the pSE380 vector. When PA20 pSE380 was grown in EMEM at 37 °C there was little evidence of flagellar-like structures (Figure 5A). When exposed to sodium butyrate, occasionally PA20 pSE380 displayed structures similar to polar flagella (Figure 5B). The expression of pSE380::*pchE* in PA20 increased both the numbers of cells displaying surface structures and numbers of structures per cell (Figure 5C). Finally, there was a complete loss of surface structures from PA20 pSE380::*pchE* when *fliC* was deleted, confirming that the observed appendages were flagella (Figure 5D).

### 2.9. HEp-2 Cell Adhesion Assays

The *pchE*-induced expression changes in the flagellar operons were widespread, more dramatic in magnitude, and enhanced rather than repressed compared to the other regulated adhesins. Therefore, we focused our adhesion investigations on the effects of flagella on HEp-2 cell attachment. Dissecting the role of long polar fimbriae in cell adhesion will be left to a different study. The percentage of PA20 adhering to HEp-2 cells was in general slightly lower than that reported in a previous study. The difference is likely to have resulted from including an additional 3h incubation in EMEM prior to the 3h challenge period, which was necessary in those experiments testing flagellar effects on attachment. The extra passage and incubation allowed longer sodium butyrate exposure without lengthening the growth phase of the challenge bacteria. However, repetitive passage in a less robust media resulted in a lower cfu/mL in the challenge dose in this study (results not shown). The results in Figure 6 were generated using strains and test conditions similar to those used for RNA isolation to allow the correlation of expression changes with adhesion data. Plasmid over-expression of *pchE* significantly reduced (*p* < 0.05) PA20 adhesion to HEp-2 cells but *pchE* deletion did not affect the percentage of cell-bound bacteria. When exposed to SMX-TM, the percentage of PA20 attached to the cultured cells was nearly five times greater (*p* < 0.05) than for PA20 in EMEM without SMX-TM. SMX-TM has been shown to greatly increase the expression of the class 1 *perC* homologues *pchA*, *pchB*, and *pch*C, resulting in strong activation of *ler* and *eae* [42]. This probably explains the large SMX-TM-induced adhesion increases seen here. The observed decreases in HEp-2 cell adhesion in the presence of *pchE* over-expression agree with earlier findings.

The results of experiments testing the effect of flagella, stimulated by butyrate exposure or induced by plasmid over-expression of *pchE*, on HEp-2 cell adhesion are shown in Figure 7. When flagellar expression was augmented by sodium butyrate exposure, the percentage of adhered cells was significantly reduced (*p* < 0.05; Figure 7A). Following *fliC* deletion, adhesion was increased nearly twofold (*p* < 0.05) suggesting that flagellar expression inhibits HEp-2 cell adhesion under these conditions (Figure 7A). In the trial shown in Figure 7B, the over-expression of *pchE* reduced the percentage of attached PA20 > 10-fold. When *pchE* was over-expressed in PA20 with deletion of *fliC* (PA20Δ*fliC*), adhesion remained at levels similar to those observed for wild-type PA20 confirming that *fliC* mediated the reductions imposed by *pchE* over-expression (Figure 7B). It should also be noted that *fliC* deletion in PA20 caused an increase in the percentage adhesion to HEp-2 cells in Figure 7A, but when *pchE* was over-expressed in the *fliC*-deleted PA20 strain (Figure 7B), adhesion dropped back to wild-type PA20 levels. This suggests that *pchE* is repressing an additional adhesin, possibly CsgA, as previously described.

## 3. Discussion

Our earlier studies showed that the expression of *pchE* reduced PA20 attachment to HEp-2 cells by repressing curli under conditions where the SOS response was not stimulated [38]. The findings also indicated that *pchE* controlled additional adhesins involved in cell attachment. The purpose of this study was to identify additional *pchE*-regulated adhesins. In that initial study, PA20 expressing pSE380 was used as a control strain to eliminate any plasmid contributions to the observed *pchE*-induced gene expression and HEp-2 cell attachment differences. Those results confirmed that *pchE* alone could generate significant changes in adhesin gene expression and in cultured cell attachment percentages. In this study, maintaining pSE380 in the control, *pchE* deleted, and SOS-induced strains of PA20 would have required the use of ampicillin, which might have interfered with SOS induction using SMX-TM. Therefore, plasmid pSE380 was not maintained in strains other than PA20 pSE380::*pchE* in the qRT-PCR comparisons. Rather, we used the PA20 pSE380 control in certain flagellar-dependent motility assays and in the SEM studies to prove that *pchE* alone would produce phenotypes expected with any newly identified gene expression changes.

Using qRT-PCR, *fliC* was strongly regulated and *lpfA1* and *eae* were modestly altered by *pchE* over-expression. The *lpfA1* and *eae* genes were repressed, similar to *csgA* in our earlier study, but *fliC* was strongly induced [38]. Surprisingly, under the conditions tested, *pchE*-induced flagella reduced PA20 attachment to HEp-2 cells. The effect of flagella on O157:H7 adhesion, as reported in the literature, depends on the cell type tested. Mahajan et al. [33] demonstrated that O157:H7 flagella are a major attachment factor for terminal rectal epithelial cells (TRE), the principal colonization site in the bovine reservoir. During adherence to bovine TRE, flagella were highly expressed in adherent bacteria at 1 h post-exposure and repressed by 3 h post-exposure [33]. However, testing Caco-2 cells, Tobe et al. [43] found that flagella were not expressed in Dulbecco’s Modified Eagle’s Medium or involved in attachments. Exposure to butyrate increased flagellar expression, and at a higher concentration elevated the expression of LEE genes to initiate intimate attachment. Activated genes within LEE, such as *grlA*, eventually repressed flagellar expression. The authors concluded that flagella are not involved in adhesion but direct EHEC to the mucosal surface by activating motility early in the infection before being repressed to minimize immune recognition during intimin-mediated attachment. Decreases in cell attachment related to flagella were also described in a study of the coordinated control of LEE and motility by Ler regulator, *grlRA* [44]. In that study, FlhDC over-expression eliminated EHEC adhesion to HeLa cells and the authors concluded that the flagellar regulon is important for controlling adhesion to host cells. In agreement with the Tobe study [43], we found that PA20 grown in EMEM did not express flagella until exposed to butyrate. However, we did observe a small but significant decrease in the cell attachment of butyrate-exposed PA20. Flagellar production was strongly increased following *pchE* over-expression and resulted in marked reductions in cell attachment that were *fliC*-dependant. A likely common mechanism explaining the reductions in cell attachment induced by both butyrate and *pchE* would be flagellar interference with a tissue adhesin expressed later in the infection process, such as intimin. However, more studies will be needed to confirm that.

PchE controls curli in part by a strong repressive effect on the curli master regulator, *csgD* [38]. We found here that PchE also affects the expression of the flagellar master regulator genes *flhDC*, probably mediating some, if not all, of the flagellar changes. When we tested additional master regulators controlling adhesive and virulence phenotypes, we observed additional changes indicating that *pchE* may impose a broad regulatory effect by acting on different global or master regulatory proteins. H-NS, a known *csgD* repressor, was unaffected by either *pchE* deletion or over-expression, but over-expression did generate a > 2-fold increase in *ler* expression, which would favor the expression of LEE and other virulence genes [40]. Moreover, *pchE* over-expression resulted in *stpA* repression. The H-NS related protein StpA is able to substitute for H-NS in the regulation of certain target genes when expressed in high concentrations, although its effect on virulence genes has not been defined [45]. Therefore, strong expression of *pchE* would enhance the Ler-mediated expression of virulence genes by increasing *ler* and possibly also by reducing *stpA* expression. An exception would be *eae*, which as noted earlier, was decreased by *pchE* expression along with certain other adhesion factors. Apparently, *pchE* exerts direct regulation of *eae* in addition to that mediated indirectly through Ler. Such a regulatory scheme would allow *ler* to control other LEE genes without increasing adhesion.

The activation of LEE and intimate cell attachment are well described properties of group 1 *perC* homologues, but the group 2 (*pchD*) and 3 (*pchE*) genes have been less well characterized [13,14,44]. This and previous studies have indicated that *pchE* functions in opposition to class 1 genes to manage cellular adhesion, likely in a temporal manner where *pchE* balances general cell adherence and motility prior to LEE activation and the initiation of stronger intimate contacts [38,42]. Earlier, we showed that PA20 expresses a low steady-state level of *pchE* that is reduced when prophage elements are induced [42]. When combined with the findings here, it appears that *pchE* evolved to support flagellar motility and minimize cell attachments until their entry into stationary growth phase or until induction of the SOS response reduces its expression, dropping flagellar production and lifting the repression on cell adhesins. Simultaneously, SOS activation of group 1 members would initiate intimate cell adhesion and further repress flagellar motility [43].

The host conditions that activate the native *pchE* promoter have not yet been defined and this needs to be considered when interpreting the significance of plasmid-induced *pchE* repression on adhesion in this study. During natural infections, regulatory control of flagellar expression has probably evolved to inhibit inappropriate or detrimental expression. More studies are needed to determine the conditions under which the *pchE* promoter might be activated and their temporal relationship with the infection process. However, this study confirms that *pchE* is a strong enhancer of flagella and could have major effects on motility and virulence. Moreover, flagella do play a role in attachment to TRE in the reservoir host, although one study indicates that their complete role is not yet fully understood [46]. The strong expression of flagella, which is detrimental for cell attachment and disease progression when inappropriately timed in the human host, could increase attachment in the reservoir host.

In this study, we identified additional EHEC attachment factors controlled by *pchE*. All the identified adhesins were repressed by *pchE* except for flagella; however, strong *pchE*-induced flagellar expression reduced rather than enhanced HEp-2 cell adhesion. Therefore, *pchE* had a general repressive effect on attachment to cultured cells, opposing the effect of class 1 regulators. Additional studies identifying natural conditions under which *pchE* is upregulated are still needed to fully understand its regulatory role in EHEC. However, the ability of *pchE* over-expression to reduce attachment to host cells and biofilm formation in the environment makes it an attractive target for intervention studies.

## 4. Materials and Methods

### 4.1. Strains, Growth Conditions, and Molecular Biology Techniques

*E. coli* serotype O157:H7 strain PA20 is a clinical isolate that carries both Stx1 and Stx2 [47]. Strains were tested and maintained in Luria Bertani (LB) broth (Miller formulation) or on LB agar, unless otherwise indicated. NEB 5-alpha (New England Biolabs, Ipswich, MA, USA) and One Shot PIR1 (Invitrogen, Carlsbad, CA, USA) were used as *E. coli* host strains for intermediary cloning steps. A rifampicin-resistant strain of PA20 was used to construct the *csgBA* deletion by allelic exchange using a SacB-encoded suicide vector to swap both genes for the spectinomycin cassette from plasmid pIC156 [48,49]. Strains PA20Δ*csgD* and PA20Δ*pchE* were constructed by Red/ET recombination replacing the open-reading-frame with a neomycin cassette. The cloning of *pchE* into inducible plasmid pSE380 (PA20 pSE380::*pchE*) has been described in a different study [38]. Strains of PA20 expressing *pchE* from pSE380 were grown without IPTG induction relying on copy-number and leaky expression from the *trc* promoter for protein production. Primers and strains are listed in Appendix A.

### 4.2. RNA Isolation, cDNA Preparation, and qRT-PCR

Strains tested for adhesin gene expression by qRT-PCR were inoculated into 3 mL LB broth and grown overnight (ON) at 37 °C with shaking at 180 rpm. The ON cultures were diluted 1:500 into 10 mL EMEM and incubated for 5 h at 37 °C with 180 rpm shaking. EMEM used to culture strain PA20 pSE380::*pchE* was enriched with 100 μg/mL ampicillin to maintain the plasmid, and the SOS response was induced in PA20 by including 540/108 μg/L SMX-TM at the start of the incubation (PA20 SMX-TM). Every strain was tested in triplicate. The samples were collected by centrifugation (4696 xg for 10 min) and the pellets were resuspended in 1 mL of RNAzol RT (Molecular Research Center, Cincinnati, OH, USA). All samples were vortexed at maximum speed for 30 s, held at room temperature for 15 min, heated for 10 min on a 55 °C heating block, and stored at −80 °C for further processing. Total RNA was extracted by the manufacturer’s protocol and cDNA was prepared with a High-Capacity cDNA Reverse Transcriptase Kit (Thermo Fisher Scientific, Waltham, MA, USA) as described previously [38].

The *pchE* RNA transcripts used for the 5′ rapid amplification of cDNA ends (5′ RACE) procedure were extracted from PA20 grown ON in LB broth in a Bactrox Hypoxia Chamber (Shel Lab, Cornelius, OR, USA) at 37 °C in 2% O_2_ and 5% CO_2_. The ON cultures were diluted 1:10 in EMEM and exposed to HEp-2 monolayers for 3 h and 5 h in the same microaerophilic chamber. Non-adherent bacteria were removed and the cells were lysed with ice-cold sterile water and centrifuged at 4 °C to remove mammalian cell debris. The supernatants were centrifuged again at 8000× *g* for 5 min at 4 °C and the pellets resuspended in RNAzol RT. Total RNA was isolated as described above.

qRT-PCR was performed by adding 1 μL (20 ng) of cDNA or DNase-treated RNA (negative control) to 19 μL reaction mixture consisting of 0.5 μM of each primer (Integrated DNA Technologies) and 10 μL Fast SYBR Green Master Mix (Applied Biosystems Foster City, CA, USA). Amplification was carried out on a 7500 Fast Real-Time PCR System (Applied Biosystems Foster City, CA, USA) using the following parameters: denaturation at 95 °C for 20 s followed by 40 cycles (3 s at 95 °C, 30 s at 60 °C). Melt curve analysis was used to verify the specificity of the amplification products. The *gyrA* gene was used as a reference to normalize the results and PA20 was used as calibrator strain to estimate the fold-change in expression for tested genes using the FC = 2-ΔΔCT method [50]. The mean FC for three trials of qRT-PCR for the selected genes and SD were reported. Comparisons with FC ≥ 2 were classified as differentially expressed.

### 4.3. Swimming Motility Assays

Swimming motility assays were performed as described by Tobe et al. [43] with slight modification. Tested strains were cultured at 37 °C without shaking for 8 h in LB containing 100 μg/mL ampicillin, if needed, for plasmid maintenance. Five-μL samples were spotted on EMEM media hardened with 0.3% agar and containing 100 μg/mL ampicillin and 20 mM sodium butyrate as indicated.

### 4.4. PchE Transcriptional Start Identification and Promoter Fusion Construction

From RNA-Seq data compiled in a previous study [42], *pchE* transcripts were aligned and the 5′ termini clustered around two specific regions 110–115 and 675–680 nucleotides upstream of the *pchE* start. The origin of the longer 675–680 nucleotide transcripts mapped to the intergenic region between the divergent ECs1585 and ECs1586 genes (Sakai annotation, NC_002695.1) (Figure 8). RNA samples were harvested from PA20 cultured on HEp-2 monolayers for both 3 h and 5 h exposures at 37 °C in 2% O_2_ and 5% CO_2_ to mimic in vivo conditions. The 5′ ends of the *pchE* transcripts were determined by RACE using the SMARTer^®^ RACE 5′/3′ kit (Takara Bio, Kusatsu, Shiga, Japan) and the primers pchE_GSP1And pchE_GSP2. A unique 650–700 bp DNA band was amplified and sequenced. This sequencing indicated that G, 10 nucleotides upstream of the ECs1586 start, corresponded to the consensus 5′ terminus of a polycistronic RNA encoding at least *pchE* and the upstream ORFs, ECs1586 and ECs1587. No shorter transcripts of 110 to 115 nucleotides were identified. As both transcript mapping and the RACE procedure placed the *pchE* transcriptional start 675 to 700 nucleotides upstream of *pchE*, the *pchE* start codon with 1067 nucleotides upstream was amplified from strain PA20 with primers 15881034F and 15881034R, restricted with *Xma*I/*Bam*HI, and cloned into the respective sites on plasmid pMLB1034 to construct a translational fusion with *lacZ* designated pMLB1034::*pchE*.

### 4.5. β-Galactosidase Assays

PA20Δ*pchE* transformed with pMLB1034::*pchE* was grown in LB broth for 18 h with shaking at 180 rpm. Overnight cultures were diluted 1:100 into 20 mL of MEM (without phenol red dye) containing 1x GlutaMAX (Gibco, Gaithersburg, MD, USA), 10 mM Hepes buffer, and 100 μg/mL ampicillin, and incubated at 37 °C with 200 rpm shaking. When the samples approached OD_600_ ≈ 0.15, each culture was divided into three 10 mL samples that were exposed either to sodium butyrate at a final concentration of 20 mM or to an equal volume of H_2_O (control samples). Aliquots of 200 μL, taken immediately (timepoint 0) or at each designated timepoint, were flash frozen in an ethanol/dry ice bath and stored at −70 °C. The OD_600_ of each culture was recorded following each sample collection. Post-collection, all samples were thawed in water and assayed for β-galactosidase activity by the procedure of Miller [51].

### 4.6. Electron Microscopy

Strains subjected to SEM were grown for 18 h in LB broth at 37 °C with shaking at 180 rpm, diluted 1:50 into EMEM with 10 mM Hepes buffer, 100 μg/mL ampicillin, and 20 mM sodium butyrate (where indicated), and incubated at 37 °C with 200 rpm shaking for 3 h. Each culture was transferred 1:50 into fresh 5 mL aliquots of the same media and incubated an additional 3 h. Samples were pelleted (Sorvall Legend X1R, TX-400 rotor) at 4696× *g* for 4 min, resuspended in 50 μL PBS, and spread on 12 mm glass slides. Samples were incubated 20 min before the wells were flooded with 1 mL 2.5% glutaraldehyde and allowed to fix for 30 min. The samples were then rinsed twice for 30 min each with 2–3 mL of 0.1 M imidazole, followed by, at 30 min intervals, 2–3 mL of each of 50%, 80%, and 90% ethanol. The samples were then washed 3 times with 2 mL of 100% ethanol and critical-point dried using liquid carbon dioxide for approximately 20 min (Denton Vacuum, Inc, Cherry Hill, NJ, USA). The samples were mounted on stubs and sputter gold coated for 1 min (EMS 150R ES, EM Sciences, Hatfield, PA, USA). Samples were then viewed with a FEI Quanta 200 F scanning electron microscope (Hillsboro, OR, USA) with an accelerating voltage of 10 KV in high vacuum mode.

### 4.7. Adhesion Assays

The adhesion of selected bacterial strains to cultured HEp-2 cells (ATCC CCL-23), adherent human epithelial cells derived from HeLa cell contamination (cervical adenocarcinoma), was compared as previously described [38]. HEp-2 cells were grown in EMEM supplemented with 10% fetal bovine serum (FBS) in a humidified 5% CO_2_ atmosphere at 37 °C. For adhesion assays, 500 µL of 1 × 10^5^ cells/mL were individually seeded into wells of tissue culture-treated polystyrene 24-well plates and cultivated until an 85%–95% confluent monolayer was formed.

Bacterial challenge strains requiring the maintenance of control or recombinant plasmids were incubated in 100 µg/mL ampicillin at each step and those strains tested for the effect of butyrate were grown in 20 mM sodium butyrate for each step beyond the initial starter cultures. FBS was not included in the EMEM used for amplifying bacteria or during the bacterial challenge of HEp-2 cells. All strains were inoculated in LB, with ampicillin if needed, and cultured at 37 °C with 180 rpm shaking for 18 h. Starter cultures (unique for each tested well) were diluted 1:100 into EMEM containing 10 mM Hepes buffer and with ampicillin and/or butyrate as needed. All strains were incubated for 3 h at 37 °C with 200 rpm shaking. Bacteria in 3 h EMEM cultures were diluted 1:10 in EMEM/Hepes containing ampicillin and sodium butyrate (as needed). The EMEM covering HEp-2 monolayers was replaced with 1 mL/well of EMEM bacterial suspension and the cells were incubated at 37 °C with 5% CO_2_ for 3 h. Afterwards, wells were individually washed 4 times with PBS to remove unbound bacteria and infected monolayers were treated with 1 mL of 0.1% Triton X-100 for 30 min. The numbers of cell-associated and wash-solution bacteria were enumerated by plate count assay and the results were reported as the percentages of attached bacteria as described [38]. In each of the three experiments, 3 or 4 independent samples of each strain were tested in individual wells. Two trials of each experiment were performed on different days and statistically analyzed as stated below.

### 4.8. Statistical Analyses

For each cell attachment experiment, a one-way negative binomial analysis of variance (ANOVA) model was fit to the unattached CFU counts observed on 3–4 independent samples for each adhesion assay, in each of three experiments; each full experiment was replicated in two independent trials. Each model produced a predicted value and 95% confidence interval for each observed unattached CFU count. These estimates were expressed as rates by dividing attached CFU by total CFU and then multiplying this value by 100%. These rates were averaged to obtain a predicted rate and 95% confidence limits for each strain. All statistical analyses were accomplished using the following R packages: MASS, ciTools, and multcomp.

## Figures and Tables

**Figure 1 ijms-21-04592-f001:**
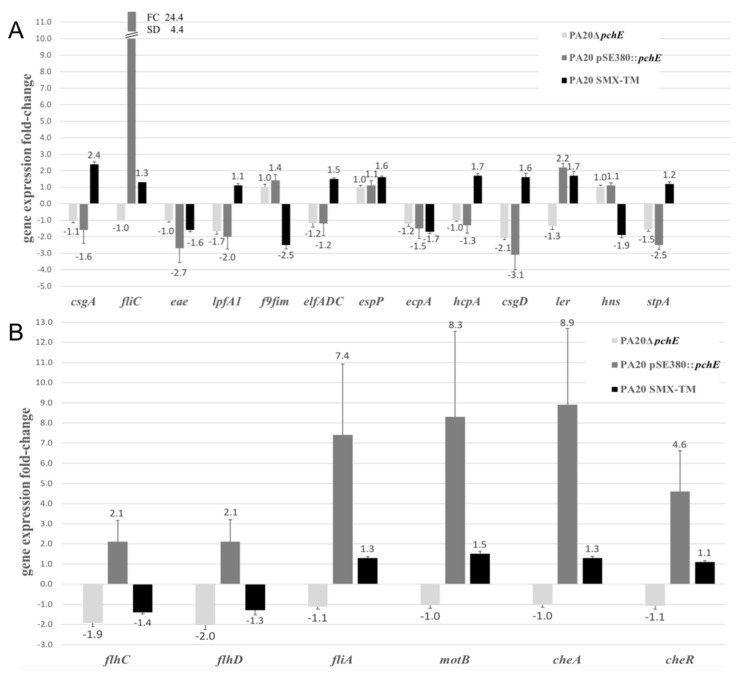
qRT-PCR-derived fold-change (FC) expression differences. Expression differences for the indicated genes derived from the comparison of strain PA20 with: 1) PA20Δ*pchE*, 2) PA20 pSE380::*pchE*, and 3) PA20 SMX-TM (PA20 exposed to 540/108 μg/L sulfamethoxazole/trimethoprim (SMX-TM)). (**A**) Genes of proven adhesins and global adhesin regulators. (**B**) Flagellar motility-associated genes. The mean FC ± SD of three trials of qRT-PCR for each indicated gene is shown.

**Figure 2 ijms-21-04592-f002:**
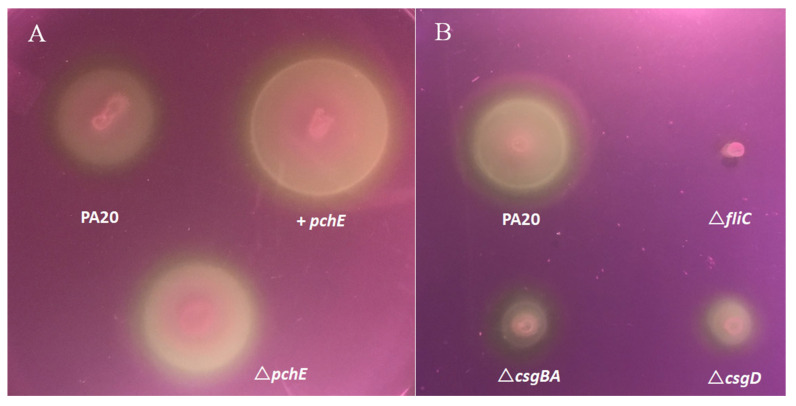
Strain swimming motility in EMEM soft agar incubated for 11 h at 37 °C. (**A**) Comparison of strains PA20 pSE380 (PA20), PA20Δ*pchE* (Δ*pchE*)*,* and PA20 pSE380::*pchE* (+*pchE*) spotted on soft agar. (**B**) Comparison of strains PA20, PA20Δ*fliC* (Δ*fliC*), PA20Δ*csgBA* (Δ*csgBA*), and PA20Δ*csgD* (Δcsg*D*) spotted on soft agar.

**Figure 3 ijms-21-04592-f003:**
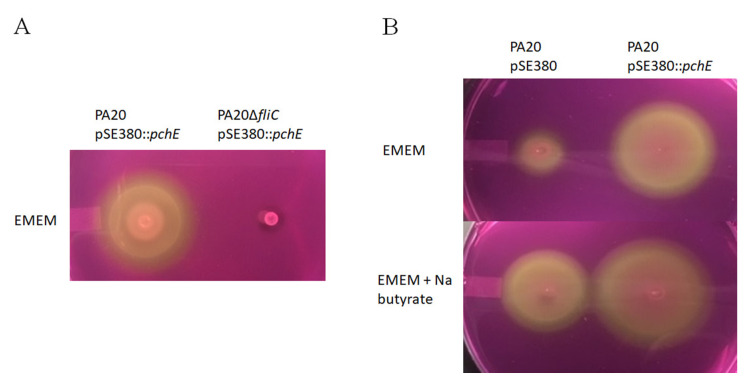
Strain swimming motility in EMEM soft agar incubated for 11 h at 37 °C. (**A**) The effect of *fliC* on PA20 swimming motility induced by *pchE.* (**B**) The effect of 20 mM sodium butyrate on strain motility.

**Figure 4 ijms-21-04592-f004:**
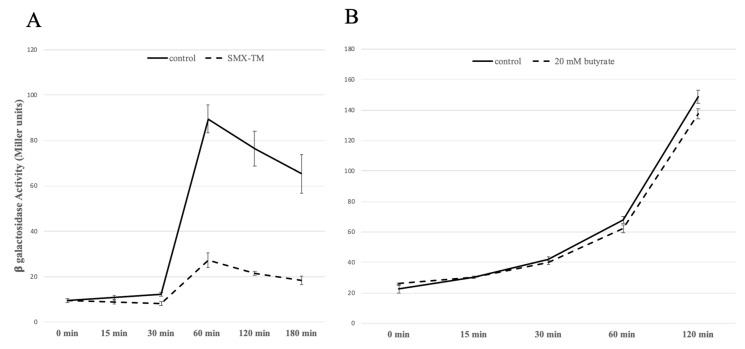
Comparisons of *pchE* promoter activities determined by β-galactosidase assays (Miller units). Growth of strain PA20Δ*pchE* pMLB1034::*pchE* in Minimal Essential Medium (MEM) at 37 °C compared with growth in MEM containing either (**A**) 540/108 μg/L SMX-TM or (**B**) 20 mM sodium butyrate for the indicated times. Each timepoint represents the sample mean and SD of four independent cultures.

**Figure 5 ijms-21-04592-f005:**
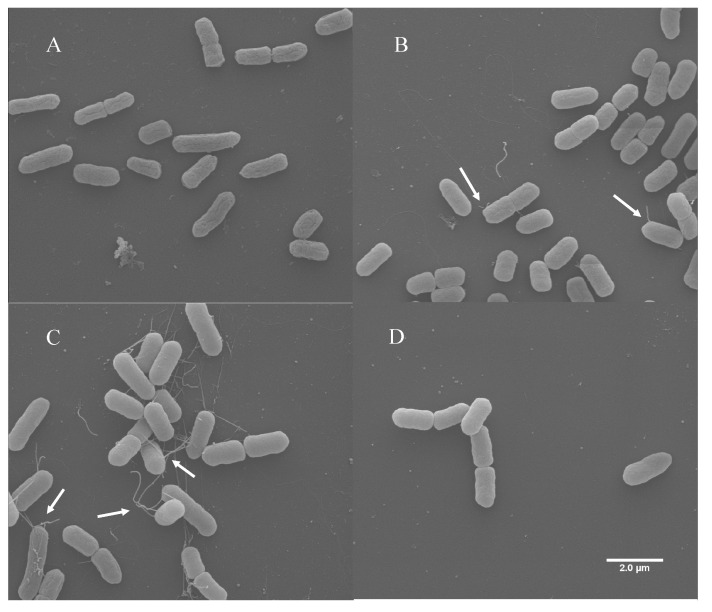
Scanning electron microscopy images. (**A**) PA20 pSE380; (**B**) PA20 pSE380 exposed to 20 mM sodium butyrate; (**C**) PA20 pSE380::*pchE*; and (**D**) PA20Δ*fliC* pSE380::*pchE* grown in EMEM at 37 °C. The arrows indicate flagella.

**Figure 6 ijms-21-04592-f006:**
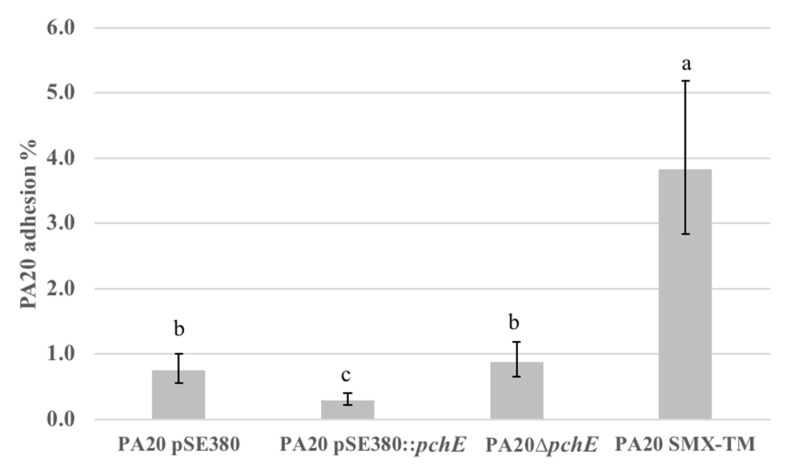
The effect of *pchE* and SMX-TM on strain PA20 adhesion to cultured HEp-2 cells. The calculated mean percentage values (and 95% confidence intervals) of two trials with three independent samples of each of adhered PA20 pSE380, PA20 pSE380::*pchE*, PA20Δ*pchE*, and PA20 exposed to 540/208 μg/L SMX-TM (PA20 SMX-TM) were determined following 3 h exposure to HEp-2 cells in EMEM at 37 **°**C in 5% CO_2_. Any two treatment means with no letter in common are significantly different (*p* < 0.05).

**Figure 7 ijms-21-04592-f007:**
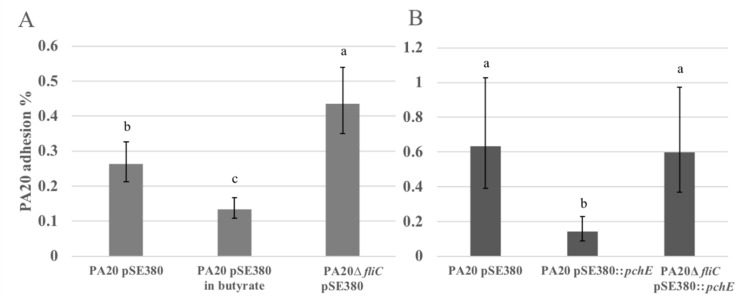
PA20 adhesion to cultured HEp-2 cells. (**A**) The effect of *fliC* and 20 mM sodium butyrate on PA20 adhesion to HEp-2 cells. (**B**) The role of *fliC* in *pchE*-induced repression of PA20 adhesion to HEp-2 cells. The calculated mean percentage values (and 95% confidence levels) of the adhered strains were determined following 3 h exposure to HEp-2 cells in EMEM at 37 °C in 5% CO_2_. Three independent samples (**A**) and four independent samples (**B**) were tested in two separate trials. Any two treatment means with no letter in common are significantly different (*p* < 0.05).

**Figure 8 ijms-21-04592-f008:**
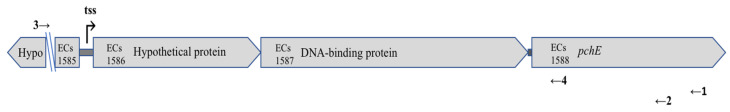
DNA map of O157:H7 PA20 *pchE* and promoter sequences showing binding sites for relevant primers used in this study. Positions and designations of *pchE* and upstream genes are annotated using the *E. coli* O157:H7 Sakai reference strain (accession #NC_002695.1). Numbers 1 and 2 designate gene-specific primers, pchE_GSP1 and pchE_GSP2, used for mapping the PA20 *pchE* transcriptional start site (tss). Numbers 3 and 4 designate primers 15881034F and 15881034R, respectively, used to amplify the DNA region 5′ to *pchE* for incorporation with *lacZ* in plasmid pMLB1034::*pchE*.

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
