# Peer review of "PchE Regulation of *Escherichia coli* O157:H7 Flagella, Controlling the Transition to Host Cell Attachment"

_ijms, 2020, doi:10.3390/ijms21134592_

Round 1

Reviewer 1 Report

The number of experimental samples and the number of repetitions should be stated in Fig.

Questions:

1.The number of experimental samples and the number of repetitions should be stated in Fig.

(It is needed at least 3 independent trials and 3 repetitions)

2. Discussion: (Lane8)In this study,maintaining...........have required the use of Ampicillin.................

Q: Canthe authors explain more about the relationship between PchE and antibiotics

3. Can the author use a cartoon to illustrate the possible mechanism?

Author Response

Reviewer 1 comments:

Questions:

1.The number of experimental samples and the number of repetitions should be stated in Fig.

(It is needed at least 3 independent trials and 3 repetitions).

Response: We included that information in the Figures (and in the Materials and Methods) as requested. For the continuous culture sampling experiments (LacZ studies), Figure 4, we sampled 4 independent cultures and plotted a time-course. For adhesion assays, Figures 6 and 7, we added the independent samples and trial numbers used for statistical analysis. For qRT-PCR in Figure 1, we added trial numbers. Note that we completed only 2 full trials of each adhesion experiment before we were shut down for the COVID19 outbreak (our labs remain closed at this time). While we would have normally performed at least three trials of each experiment, the data in two trials was sufficient to perform the statistical analysis we describe. Moreover, we believe that the strong phenotypic data shown here (in motility assays, SEM, and butyrate trials) and the dramatic changes in expression of flagellar operons will adequately support the findings in the adhesion assays.

  1. Discussion: (Lane8) In this study, maintaining...........have required the use of Ampicillin.................

Q: Can the authors explain more about the relationship between PchE and antibiotics

Response: There is no special relationship regarding pchE and antibiotics (other than that antibiotics have been shown to induce the SOS response, which is stated and referenced in the first sentence of the Discussion). The plasmid (pSE380) used to express pchE in this study encodes the ampicillin gene and ampicillin is needed only to assure maintenance of pSE380 in transformed strains. The explanation starting on line 8 is given only to explain why pSE380 could not be included in the other strains used in the RT-PCR comparisons. We re-wrote this section and hopefully that will make it less confusing for readers without expanding the text.

  1. Can the author use a cartoon to illustrate the possible mechanism?

Response: The reviewer is likely referring to the mechanism of the antibiotic relationship with pchE. As explained in the previous point, the use of antibiotic is for plasmid maintenance only, not for an interaction/effect on pchE. If the reviewer is asking for an explanation of the possible mechanism of action of pchE on adhesin or flagellar genes, the answer is that it is unknown at this time. Other more characterized members of the pch family are transcriptional regulators and it is likely that pchE uses the same mechanism. However, studies of pch protein interactions with promoters of affected genes have not yet been performed.

Reviewer 2 Report

Overall the paper is well written and is from an experienced research group exploring pathogenicity of an important pathogen of public health concern that has very important regulatory status in the food commerce as well. I would recommend this article for publication after undergoing certain revisions.

Nomenclature of pathogenic Escherichia coli is extremely important and a moving target. In reviewer’s opinion this could be better illustrated in the introduction. It is important, the reviewer believes, to discuss correctly the pathotypes, serogroups, serovars, and categories of the intestinally pathogenic E. coli that could encode Shiga toxin. It is also essential to this discuss that these boundaries are blared due to the 2011 outbreak of E. coli O104 in Germany were one strain exhibited characteristics of EHEC and EAEC. As a starting point an including some references from CDC’s MMWR and Nature Reviews. This information could be expanded in the introduction and/or articulated in discussions.

Kaper, J.B., Nataro, J.P. and Mobley, H.L., 2004. Pathogenic escherichia coli. Nature reviews microbiology2(2), pp.123-140.

Foley, C., Harvey, E., Bidol, S.A., Henderson, T., Njord, R., DeSalvo, T., Haupt, T., Mba-Jonas, A., Bailey, C., Bopp, C. and Bosch, S.A., 2013. Outbreak of Escherichia coli O104: H4 infections associated with sprout consumption—Europe and North America, May–July 2011. MMWR. Morbidity and mortality weekly report62(50), p.1029.

Statistics are not used correctly. In Figures 5 and 6 as an example, typically the largest value received the “a.” This is the standard output of R packages. Design is unclear as well. This is extremely important to assure a study is repeatable and has internal validity. How were the trials replicated? It is common to perform assays in biologically independent repetitions as blocking factors of a randomized bock design and then within each block there is need to have replications based on power and sample size analyses. It is also common to have instrumental replication to assure a measurement is precise. These pieces of information would need to be incorporated in the Materials and Methods section. Assays are explained well, design and replication pieces of information are missing.

Figure one is practically impossible to read. It could be divided into smaller graphs and then merged as panels of a large figure or re-structured. The article have several figures and tables, although all relevant to discussion, some could be transferred to supplement. Specially table one that is extended in more than one page. One other alternative is to use free and public repositories, like Harvard Dataverse, that is easy to use and free, and then provide a DOI link to the readership to obtain the info, if they need, by clicking on the link.

Section 4.6, authors indicate they harvest the cells at “5000 xg for 4 min.” Would it be possible to check this number to assure it is correct. Centrifuges operated based on RPM that is not a metric system so authors would need to calculation g force used to harvest the cells by converting RPM to g force. An important characteristic is the router dimension of the unit so the name of the unit and the router model and dimention would also need to be provided. Then a simple calculation could assure the correct g force is reported. Here is an online tool from a research supplier as an example:

The provided value may or may not be correct, it would make sense to name the unit, the router and re-calculate the reported value to assure correctness.

Author Response

Reviewer 2 comments:

Comments and Suggestions for Authors

Overall the paper is well written and is from an experienced research group exploring pathogenicity of an important pathogen of public health concern that has very important regulatory status in the food commerce as well. I would recommend this article for publication after undergoing certain revisions.

Nomenclature of pathogenic Escherichia coli is extremely important and a moving target. In reviewer’s opinion this could be better illustrated in the introduction. It is important, the reviewer believes, to discuss correctly the pathotypes, serogroups, serovars, and categories of the intestinally pathogenic E. coli that could encode Shiga toxin. It is also essential to this discuss that these boundaries are blared due to the 2011 outbreak of E. coli O104 in Germany were one strain exhibited characteristics of EHEC and EAEC. As a starting point an including some references from CDC’s MMWR and Nature Reviews. This information could be expanded in the introduction and/or articulated in discussions.

Kaper, J.B., Nataro, J.P. and Mobley, H.L., 2004. Pathogenic escherichia coli. Nature reviews microbiology2(2), pp.123-140.

Foley, C., Harvey, E., Bidol, S.A., Henderson, T., Njord, R., DeSalvo, T., Haupt, T., Mba-Jonas, A., Bailey, C., Bopp, C. and Bosch, S.A., 2013. Outbreak of Escherichia coli O104: H4 infections associated with sprout consumption—Europe and North America, May–July 2011. MMWR. Morbidity and mortality weekly report62(50), p.1029.

Response: We agree that the diversity of and the nomenclature for E. coli pathogens is extremely important. However, it is a huge subject and one that is extremely hard to summarize in a paragraph or two. We worked some of the requested information into the very first paragraph of the Introduction, focusing only on the intestinal pathotypes. We focused more on pathotypes and gave references so that readers could easily find more information on specific serotypes in each group. We also tried to focus our comments so that it tied directly into genes important in cell adhesion and biofilms in the Shiga toxin-containing strains, the subject of this study.  

Statistics are not used correctly. In Figures 5 and 6 as an example, typically the largest value received the “a.” This is the standard output of R packages. Design is unclear as well. This is extremely important to assure a study is repeatable and has internal validity. How were the trials replicated? It is common to perform assays in biologically independent repetitions as blocking factors of a randomized bock design and then within each block there is need to have replications based on power and sample size analyses. It is also common to have instrumental replication to assure a measurement is precise. These pieces of information would need to be incorporated in the Materials and Methods section. Assays are explained well, design and replication pieces of information are missing.

Response: We re-labeled Figures 6 and 7, following R convention, using “a” for the largest value as requested.

            We also added more information on experimental design, rather than just listing a reference for a past study. The number of independent trials and independent samples are now stated in the Figure captions (as requested by reviewer 1), listed also in the “Adhesion Assays” section of the Materials and Methods, and listed in the “Statistical Analysis” section of the Materials and Methods. The analysis of adhesion data with the 2 trials of data includes a “Trial” block effect testing trial x strain interactions. We briefly expanded the first sentences in the “Adhesion Assays” section of the Materials and Methods so readers know that the Trials were included in the analyses and a Trial effect was tested.

            Note that we completed only 2 full trials of each adhesion experiment before we were shut down for the COVID19 outbreak (our labs remain closed at this time). While we would have normally performed at least three trials of each experiment, the data in two trials was sufficient to perform the statistical analysis we describe. Moreover, we believe that the strong phenotypic data shown here (in motility assays, SEM, and butyrate trials) and the dramatic changes in expression of flagellar operons will adequately support the findings in the adhesion assays.

            The results of the adhesion assays were recorded as plate counts and did not use machine measurements.

Figure one is practically impossible to read. It could be divided into smaller graphs and then merged as panels of a large figure or re-structured. The article have several figures and tables, although all relevant to discussion, some could be transferred to supplement. Specially table one that is extended in more than one page. One other alternative is to use free and public repositories, like Harvard Dataverse, that is easy to use and free, and then provide a DOI link to the readership to obtain the info, if they need, by clicking on the link.

Response:

            We split Figure 1 as requested. We also modified the appearance of the fliC expression data-bar in panel (A) so that the scale of the graph was reduced to a maximum of 11-fold, which made the difference between the various bars more obvious.

            Also, when we replaced Figures 6 and 7 after re-analysis of the adhesion data, we enlarged the letters and numbers to make it easier to read.

            We labeled Table 1 and Figure 8 as Supplementary files as requested, which should reduce the manuscript size. However, we left the Table and Figure in the same place in the revised manuscript for review.

 Section 4.6, authors indicate they harvest the cells at “5000 xg for 4 min.” Would it be possible to check this number to assure it is correct. Centrifuges operated based on RPM that is not a metric system so authors would need to calculation g force used to harvest the cells by converting RPM to g force. An important characteristic is the router dimension of the unit so the name of the unit and the router model and dimention would also need to be provided. Then a simple calculation could assure the correct g force is reported. Here is an online tool from a research supplier as an example:

The provided value may or may not be correct, it would make sense to name the unit, the router and re-calculate the reported value to assure correctness.

Response: The number given was an rpm value not the g-force, which was our mistake. We  have listed the centrifuge, the rotor, and the company-published g-force in the revised manuscript in section 4.6. The published number was similar to what was predicted using the website listed by the reviewer.

Round 2

Reviewer 2 Report

There had been considerable improvements in this revised version of the manuscript and vast majority of the provided feedback had been incorporated. I would suggest minor revisions on the first two graphs prior to further consideration of this manuscript. As previously requested, figures one and two of the study do not currently have high publication quality. Values provided are practically impossible to read with naked eye and values of vertical and horizontal axes are not readable. With additional programming in R studio or using Microsoft products, figures 1 and 2 could be readable for IJMS readership. Figure 6 is improved now and could be used as a model, all the values are eligible and contents could be assimilate by stakeholders.

Author Response

Reviewer 2 comment:

There had been considerable improvements in this revised version of the manuscript and vast majority of the provided feedback had been incorporated. I would suggest minor revisions on the first two graphs prior to further consideration of this manuscript. As previously requested, figures one and two of the study do not currently have high publication quality. Values provided are practically impossible to read with naked eye and values of vertical and horizontal axes are not readable. With additional programming in R studio or using Microsoft products, figures 1 and 2 could be readable for IJMS readership. Figure 6 is improved now and could be used as a model, all the values are eligible and contents could be assimilate by stakeholders.

Response:

We have worked with the first 2 graphs (parts A and B of Figure 1) and made several additional changes. We increased the font size of the vertical axis labels, the vertical fold-change numeric values, the horizontal individual gene labels, and especially the individual fold-change numbers associated with each bar. When we tried to increase the size of these features more, we started to crowd the elements within the graph and distorted the visual presentation. If further enlargement is required, it would be necessary to break the figure into three, rather than two, parts. This might make it more confusing. With the majority of readers viewing on-line and able to adjust the view to their own needs, we hope this will be a reasonable compromise.